# Enhanced Infrared Absorbance of the CMOS Compatible Thermopile by the Subwavelength Rectangular-Hole Arrays

**DOI:** 10.3390/s20113218

**Published:** 2020-06-05

**Authors:** Chi-Feng Chen, Chih-Hsiung Shen, Yun-Ying Yeh

**Affiliations:** 1Department of Mechanical Engineering, National Central University, Taoyuan City 32001, Taiwan; ccf@cc.ncu.edu.tw (C.-F.C.); j9038592@ms38.hinet.net (Y.-Y.Y.); 2Department of Mechatronics Engineering, National Changhua University of Education, Changhua City 50007, Taiwan

**Keywords:** subwavelength, subwavelength hole arrays, subwavelength rectangular-hole arrays, thermopile, CMOS-MEMS, infrared radiation, infrared sensors, infrared absorbance

## Abstract

The enhanced infrared absorbance (IRA) of the complementary metal-oxide-semiconductor (CMOS) compatible thermopile with the subwavelength rectangular-hole arrays in active area is investigated. The finite-difference time-domain (FDTD) method considered and analyzed the matrix arrangement (MA) and staggered arrangement (SA) of subwavelength rectangular-hole arrays (SRHA). For the better cases of MA-SRHA and SA-SRHA, the geometric parameters are the same and the infrared absorption efficiency (IAE) of the SA type is better than that of the MA type by about 19.4% at target temperature of 60 °C. Three proposed thermopiles with SA-SRHA are manufactured based on the 0.35 μm 2P4M CMOS-MEMS process. The measurement results are similar to the simulation results. The IAE of the best simulation case of SA-SRHA is up to 3.3 times higher than that without structure at the target temperature of 60 °C. Obviously, the staggered rectangular-hole arrays with more appropriate geometric conditions obtained from FDTD simulation can excellently enhance the IRA of the CMOS compatible thermopile.

## 1. Introduction

In recent years, with the development of consumer electronic products and the threat of fever-type infectious diseases, such as influenza or coronavirus pneumonia, the demand for infrared (IR) sensors at room temperature has been increasing, especially in mobile and wearable device applications. This means that miniaturization and cost-reducing manufacturing technologies will be the key to the development of room IR sensors. The thermocouple is a widely used room temperature sensor [1,2,3]. Thermocouples can be connected in parallel to form a so-called thermopile, where all hot junctions are located at the higher temperature end and all cold junctions are located on the other side. Thermopile output is a voltage, which is directly proportional to the temperature difference between the cold and hot junctions. Thermopile is used to measure the intensity of incident IR radiation is so-called thermopile sensor. It is suitable for application in remote temperature sensing [4,5] and non-dispersive infrared sensing (NDIR) gas detection [6].

Micro-electro-mechanical systems (MEMS) process technology has the advantages of making light, thin, small components, and low-cost mass production [7,8,9]. It is one of the necessary technologies to integrate microelectromechanical components and integrated circuits on the same chip simultaneously in order to meet the development trend of integration and miniaturization. The so-called monolithic complementary-metal-oxide-semiconductor (CMOS) MEMS or CMOS-Compatible MEMS is a development technology that integrates both standard CMOS integrated circuits and micro-electromechanical systems [10,11,12]. It just meets the needs of sensor technology development and manufacturing and, of course, also meets that of thermopiles. A small thermopile sensor D6T that was manufactured by such MEMS technology was developed by Japan’s Omron in 2012 [13]. This sensor structure can be combined with the mobile phone image function. This means that the thermopiles manufactured by MEMS technology can meet the needs of consumer electronic products or portable devices.

In addition, improving sensor sensitivity will be one of the focuses of IR sensor research. A CMOS compatible thermopile with several subwavelength hole arrays is presented [14]. The measurement and simulation results consistently show that the infrared absorption efficiency (IAE) of the CMOS compatible thermopile has been significantly improved when there is a subwavelength hole structure in absorption area. Continuing the above research, several special subwavelength columnar structures added in rectangular hole of the best case of the above research are numerically investigated by the finite-difference time-domain method in order to enhance the IAE of the CMOS compatible thermopile [15]. It is obtained that the IAEs of the better cases for the types of three rectangular columns and three ellipse columns can be increased by 14.4% and 15.2%, respectively.

In this study, we numerically and experimentally investigate the enhanced infrared absorbance (IRA) of a CMOS compatible thermopile, adding the subwavelength rectangular-hole arrays (SRHA) in the absorption area. Two arrangements of SRHA are considered and analyzed by the finite-difference time-domain (FDTD) method [16,17,18,19], one is matrix arrangement (MA), or called square arrangement, and the other is staggered arrangement (SA), or called hexagon arrangement. It is obtained that, for the same hole parameter, the SA type is usually better than the MA type. Three simulated thermopiles with SA-SRHA are manufactured by the 0.35 μm 2P4M CMOS-MEMS process in TSMC (Taiwan Semiconductor Manufacturing Company). Finally, we complete the measurement of trial productions and compare with the simulation results. Obvious, the better case of SA type has excellent IR absorptive capacity than the previous studies.

## 2. Simulation Results and Discussion

We can clearly know that the chip structure of this process from the design rules of the 0.35 μm 2P4M CMOS-MEMS process in TSMC [14]. In this chip, SiO2 is mainly the medium and it can effectively absorb the transmission IR lightwave. Therefore, we adopt this process to fabricate our thermopile. The thermopile is designed and Figure 1 shows the sketch. The active area used to absorb IR radiation is configured in hot junction end and the cold junction is at the other end of thermocouple array. In this study, the temperature of cold junction is the ambient temperature and is about 30 °C. In the active area, two arrangements of SRHA are considered and the top-view sketches of MA-SRHA and SA-SRHA are shown in two sub-graphs of Figure 1. In the sub-graphs of Figure 1, the hole widths in the x-axis and y-axis directions are represented by *w_x_* and *w_y_*, respectively. Additionally, the hole wall width denotes the wall width between two holes and the sizes in the x-axis and y-axis directions are assumed as the same and they are represented by the symbol *ρ.* To search the better geometric parameters, an accurate and available technique, the FDTD method, is used. Furthermore, we try to consider only using SiO_2_ to represent the entire medium of active area in order to simplify the simulation model while maintaining the principle of accuracy. Figure 2 shows the sketch of the IR lightwave propagation from incident medium along z-axis direction into the active area of a thermopile with SRHA. The IR lightwave through the active area will be gradually absorbed. If there is IR lightwave that is not completely absorbed by the material, it will be transmitted to the transmission medium. The refractive indexes of incident medium, active-area medium, and transmission medium are represented by *n_0_*, *n_a_*, and *n_t_*, respectively, where we take as *n_0_* = 1, *n_a_* = 1.42, and *n_t_* = 1. Generally, when simulation technology is used to solve electromagnetic wave problems, it is impossible to set an infinite space because of computer memory and calculation speed limitations. Therefore, an artificial absorbing boundary condition, the so-called perfectly matched layer (PML), in the FDTD simulation was originally proposed by Berenger et al. [20] and it is used to effectively suppress the reflection at the analysis window [16,21]. This method involves surrounding the computing unit with a medium and, in theory, this medium can absorb electromagnetic waves of all frequencies and angles of incidence without any reflection. That is, it can reduce the error that is caused by the boundary of the limited simulation area. According to the principle of conservation of energy, the radiant energy should be absorbed by the material after subtracting the reflection part and the transmission part. One can see from Figure 2 that the reflection part and the transmission part of radiant energy will be received by the reflection detector and the transmission detector, respectively. We have counted the total absorption of active area medium and compared with the result that was calculated from the principle of conservation of energy. The results show that the two are almost identical. 

First, before using the simulation tool based on this simplified material model to design the CMOS compatible thermopiles with various SRHAs, we confirm the accuracy of this simulation tool. It is found that the simplified material model is a good approximation after comparing the simulation results of the simplified material model and the complete material model. In addition, according to previous research results [14], the agreement between the simulation and experimental results is good. Therefore, this simplified simulation tool should be used to effectively predict the IAE response of CMOS compatible thermopiles with various SRHAs.

In the previous study [14], we consider the limitation of the minimum structure line width is 3 μm according to their design rules of TSMC’S 0.35 μm 2P4M CMOS-MEMS process and take the minimum structural thickness as 3 μm. Therefore, we directly take the minimums of the hole-wall width and the hole width as 3 μm in the simulation working, and assume that the hole-wall width *ρ* and the hole width in the y-axis direction *w_y_*, are the same, that is, the *ρ = w_y_*. For the type of MA-SRHA, the *ρ* is considered as 3 μm, 3.5 μm, and 4 μm, and the *w_x_* is adjusted from 3 μm to 21 μm and the interval is 3 μm. It is obtained that the best results for IAE are the pattern of *ρ* = *w_y_* = 3.5 μm and *w_x_* = 15 μm, and the second one is that of *ρ* = *w_y_* = 3 μm and *w_x_* = 15 μm. Two patterns are used to fabricate the samples. The normalized simulation and experiment results normalized by the individual maximum are obtained and shown in Figure 3. It can be seen that the simulation results are similar to the experiment results. Here, the measured room temperature is set at 30 °C, so that, in Figure 3, two values are 0 at target temperature 30 °C. In addition, it is found that the minimum line width of 2.5 μm can be achieved, according to the previous manufacturing results of TSMC’S 0.35 μm 2P4M CMOS-MEMS process. Therefore, in this study, the minimum structural size is taken as 2.5 μm and the tested parameter range of the related geometric parameters is shown in Table 1.

Figure 4 shows the variances of the IRA with different *ρ* by the function of *w_y_* for the thermopiles with SA type (a) and MA type (b) of SRHA of *w_x_* = 15 μm. Here, IRA is calculated for the wavelength range of 8 μm to 10 μm. This is because, in this study, the target temperature is set in the range of 30 °C to 100 °C, so the calculated wavelength range is correspondingly selected in the range of 8 μm to 10 μm. It is seen that both types are the same, the maximum of IRA occurs when *ρ* = 2.5 μm and *w_y_* = 5 μm, and the IRA of SA type is about 93.76% and that of MA type is about 87.92%. 

For the same hole shape, SA type is better than MA type. Based on the above results, SA type is selected as the experimental target. Figure 5 shows the variances of the IRA with different *w_y_* by the function of *w_x_* for the thermopiles with SA type (a) and MA type (b) of SRHA. One can see that the best results are obtained when *w_x_* = 15 μm and *w_y_* = 5 μm and are just the same as that of Figure 4. We choose SA type patterns for the experiment since the SA type results are significantly better than the MA type results. Before the experiment, we further fine-tune the geometric parameters to find more appropriate geometric parameters and the results are shown in Figure 6. For three trial thermopiles with various SA type patterns, Table 1 lists the rectangular-hole sizes and their simulated IRAs. In addition, to compare the experimental results, the normalized IAEs normalized by the maximum of them at 60 °C with the target temperature for three proposed thermopiles and that without structure are used and shown in Figure 7 and listed in Table 2. The best result is about 94.39% and was obtained at *ρ* = 2.5 μm, *w_x_* = 15.5 μm, and *w_y_* = 5.5 μm.

## 3. Experimental Results and Discussion

Inheriting our existing mature micro-electromechanical process technology and verifying the sub-wavelength structure in this study, the thermopile sensor was fabricated in TSRI (Taiwan Semiconductor Research Institute) while using 0.35 μm 2P4M CMOS-MEMS process. This process is realized under a series of anisotropic etching after the TSMC standard 0.35 μm CMOS process. Because the thermopile works based on the principle of Seebeck effect, the chip was designed using the standard process of polysilicon and metal aluminum. Multiple proposed thermopile units are configured with various SA-SRHA patterns in the same chip in order to compare and analyze the effect of IR thermal radiation absorption.

Regarding the fabrication of the SRHA sensing structure, it is proceeded after the standard TSMC CMOS process, using some specific post-processing MEMS processes including two RLS and RLSSI processes provided by TSRI. After PAD opening in the CMOS process, the SRHA pattern is formed by reactive ion etching (RIE) in order to remove SiO_2_ dielectric material on the absorption region in the RLS process. Next, the RLSSI process is used to further perform RIE etching of the silicon substrate beneath under the sensing structure. After the processes, the active area is floated and filled with an array of etched holes as a periodic index waveguide. After removing the silicon substrate under the cantilever, the SiO_2_ thin film structure in the active area has a thickness of 7 μm. The design of the thermocouple is an n-type polysilicon strip with a width of 20 μm and length of 200 μm. An aluminum wire with a minimum rule of 0.5 μm width is deposited on top of n-type polysilicon. After the process is completed, the cantilever beam with poor thermal conductivity is suspended on the etching cavity, which serves as a good carrier for the investigation and analysis of the sensor structure and SRHA.

Figure 8 shows the SA-SRHA for the CMOS compatible thermopile, respectively. One can see that the structures of SA-SRHA are well fabricated for thermopile devices and both are successfully fabricated by the CMOS-MEMS process. It is worth noticing that holes of SA-SRHA are well defined as our design. The size of design, 5.50 × 15.50 μm for rectangular hole after dry etching of active area is deviated to 5.51 × 15.60 μm in Figure 8. Table 3 shows the measurement results for the geometric parameters of three proposed thermopiles, where ρ_x_ and ρ_y_ are the hole-wall widths in the x-axis and y-axis directions, respectively. The manufacture of this prototype is successful, although there are still some deviations. The geometric deviations will slightly affect the property of thermopiles. We re-execute the simulation based on those measurement geometric parameters to more objectively compare simulation and experiment. The simulation results of IAEs is shown in Figure 9. 

Subsequently, we proceed to study and measure the properties of the trial thermopiles; the experimental measurement framework is set in Figure 10. The output voltage of the SRHA thermopile is measured under various conditions of target temperatures and shown in Figure 10a. The measurement setup consists of a standard IR radiation source (IRS) and a modulated mechanical chopper system. The output signal of the thermopile is amplified by a low noise and low temperature drift chopper amplifier AD8551, and the signal is delivered to a data acquisition (DAQ) device, NI USB-6009 (National Instruments, Austin, TX, USA). The distance between the standard IRS and the trial thermopile chip is fixed at 40 mm. The temperature of the standard IRS is set to 30–60 °C with an interval of 10 °C. In addition, a low-frequency chopper is installed before the IR thermopile and the 5–14 μm IR filter in order to avoid the interference of ambient light or other background signals.

Further, the proposed thermopile is investigated for its speed of response, and the frequency response measurement setup is established for analysis in Figure 10b. Here, a controller modulates the mechanical chopper. There is a reference frequency from the controller that is delivered to a lock-in amplifier to acquire the frequency response of thermopile output voltage V_th_. The floating structure of thermopile is the same as the previous study, where the measurement time constant is similar about 4.8 ms, and the corresponding bandwidth is 32 Hz. Figure 11 shows the normalized output voltages normalized by normalized by the maximum of test 2 case for three proposed thermopiles. When comparing Figure 11 with Figure 9, it can be found that the normalized experiment results are similar to the normalized simulation results. Table 4 shows the simulation results of the normalized IAE normalized by the maximum of them at target temperature 60 °C for several thermopiles with SRHA of the given geometric parameters, the first case is the best-simulation-case based on the geometric parameters of actual manufacturing, the second case is the best-simulation-case for MA type, the third case is the best-case in [14], and the last is without SRHA. Obviously, the SA case is much better than others. At the target temperature of 60 °C, the IAE of the best case of SA-SRHA is up to 3.3 times higher than that without structure.

## 4. Conclusions

In this study, the enhanced IRA of the CMOS compatible thermopile with the MA-SRHA and SA-SRHA in active area is investigated. We analyze these geometric parameters one by one to find the geometry with better IRA by using the FDTD method. It is obtained that, for the same hole parameter, the SA type is usually better than the MA type and, for the better cases, their geometric parameters are the same, the geometric size of hole are 15.5 μm and 5.5 μm, and the hole-wall width is 2.5 μm. Based on the 0.35 μm 2P4M CMOS-MEMS process in TSMC, three proposed thermopiles with SA-SRHA are manufactured. The results of measurement and simulation are similar. At the target temperature of 60 °C, the IAE for the best-simulation-case based on the geometric parameters of actual manufacturing is 24.1% higher than that of the MA type and 46.9% higher than that of the best-case in [14]. Obviously, the staggered rectangular-hole arrays with more appropriate geometric conditions being obtained from FDTD simulation can excellently enhance the IRA of the CMOS compatible thermopile.

## Figures and Tables

**Figure 1 sensors-20-03218-f001:**
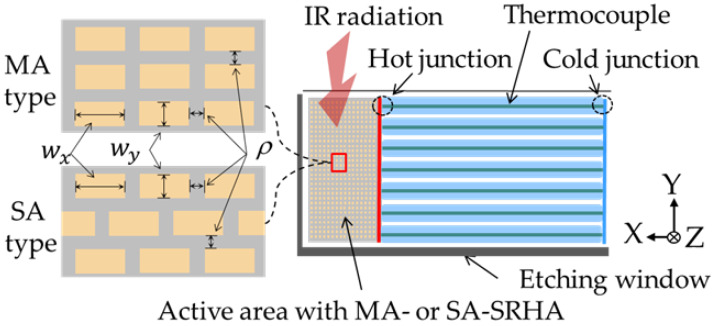
Sketch of the complementary-metal-oxide-semiconductor (CMOS) compatible thermopile and top-view sketches of matrix arrangement-subwavelength rectangular-hole arrays (MA-SRHA) and staggered arrangement-subwavelength rectangular-hole arrays (SA-SRHA) shown in two sub-graphs.

**Figure 2 sensors-20-03218-f002:**
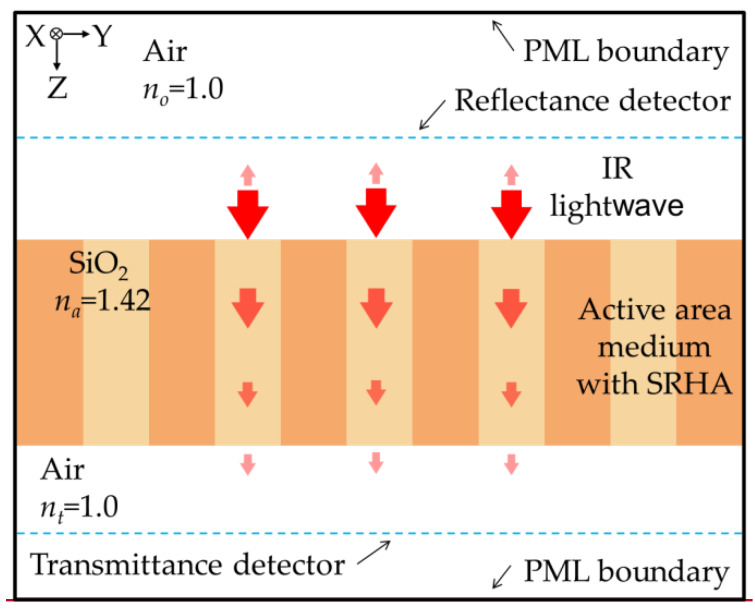
Sketch of a light-wave propagation through a CMOS compatible thermopile with SRHA simulated by the finite-difference time-domain (FDTD) method.

**Figure 3 sensors-20-03218-f003:**
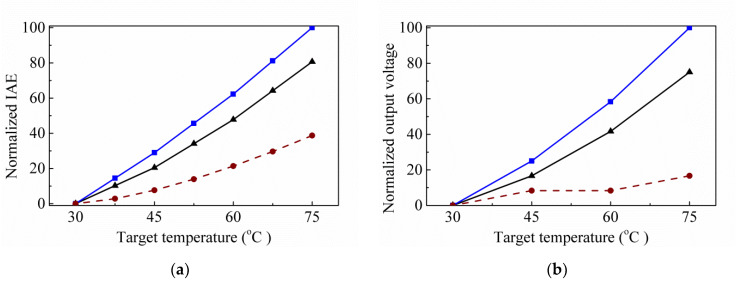
Normalized infrared absorption efficiencies (IAEs) (**a**) and normalized output voltages (**b**) normalized by the individual maximum with the target temperature for the thermopiles with MA-SRHAs and without structure, where the solid lines with triangle and square respectively represent the cases of *w_y_* = 3 μm and *w_y_* = 3.5 μm, and the dashed lines with round represents w/o structure case.

**Figure 4 sensors-20-03218-f004:**
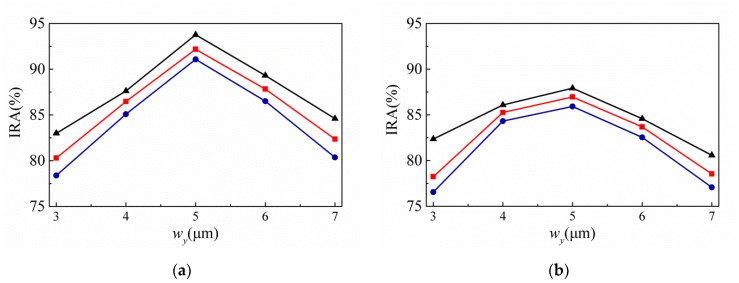
Variances of the infrared absorbance (IRA) with different *ρ* by the function of *w_y_* for the thermopiles with SA type (**a**) and MA type (**b**) of SRHA of *w_x_* = 15 μm, where the solid lines with triangle, square, and round represent the cases of *ρ* = 2.5 μm, *ρ* = 3μm, and *ρ =* 3.5 μm, respectively.

**Figure 5 sensors-20-03218-f005:**
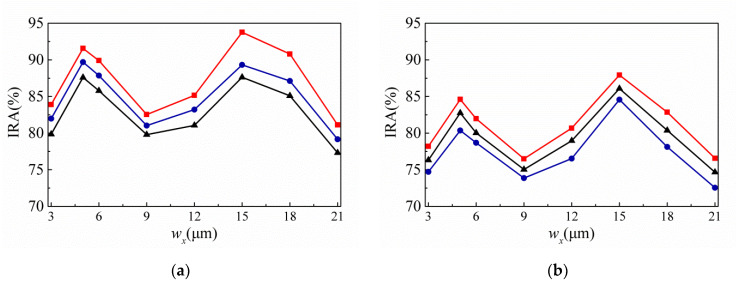
Variances of the IRA with different *w_y_* by the function of *w_x_* for the thermopiles with SA type (**a**) and MA type (**b**) of SRHA, where the solid lines with triangle, square, and round represent the cases of *w_y_* = 4 μm, *w_y_* = 5 μm, and *w_y_* = 6 μm, respectively.

**Figure 6 sensors-20-03218-f006:**
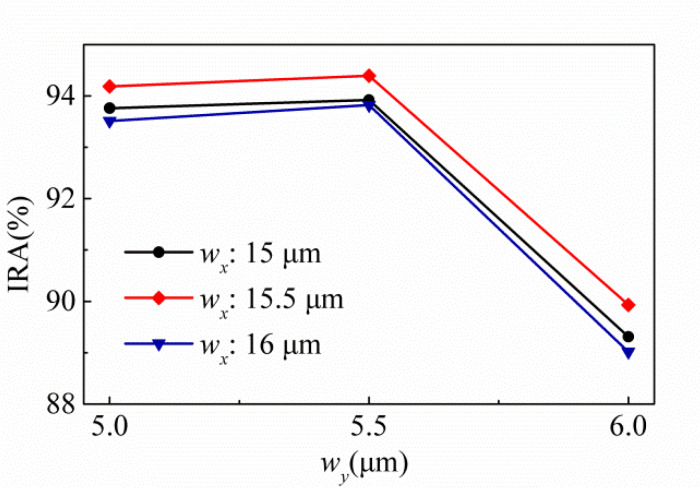
Variances of the IRA with different *w_x_* and *w_y_* for the thermopiles with various SA-SRHA *w_x_* of 15, 15.5, and 16 μm.

**Figure 7 sensors-20-03218-f007:**
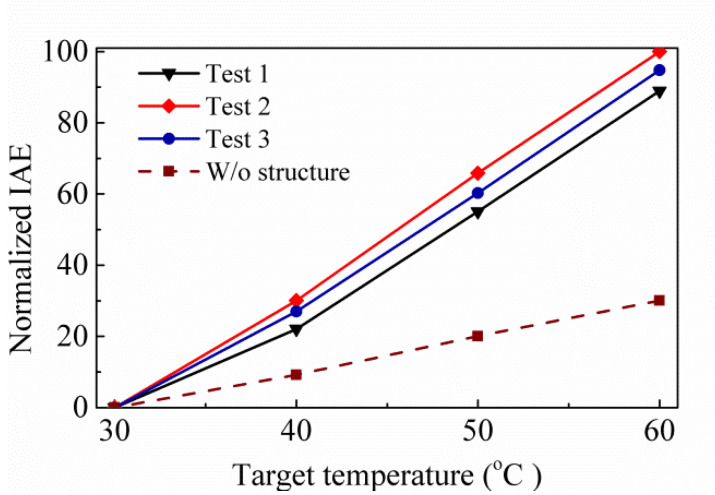
Normalized IAEs normalized by the maximum of them with the target temperature for three proposed thermopiles and that without structure.

**Figure 8 sensors-20-03218-f008:**
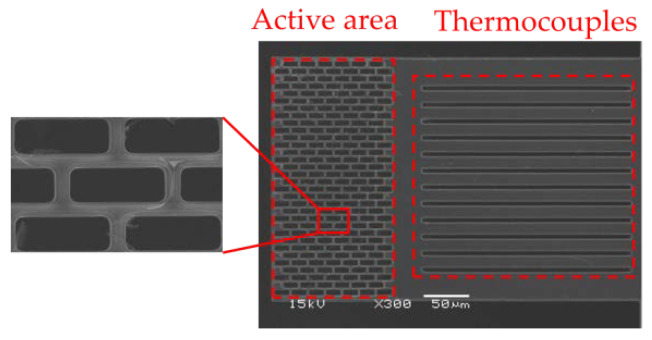
SA-SRHA SEM image of the CMOS compatible thermopile.

**Figure 9 sensors-20-03218-f009:**
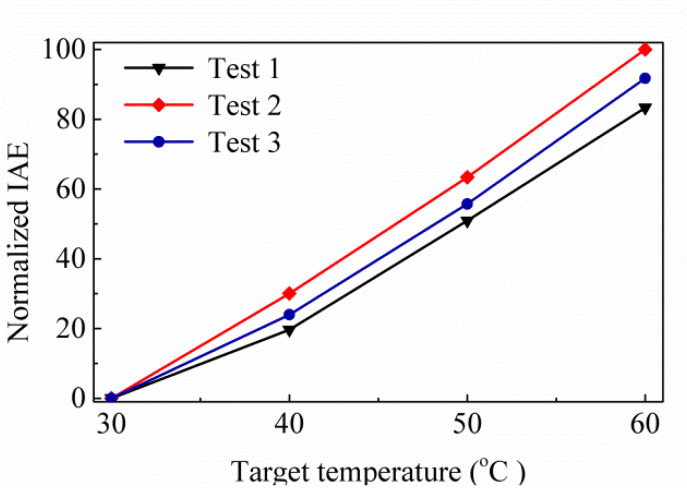
Normalized IAEs normalized by the maximum of test 2 case for three trial thermopiles.

**Figure 10 sensors-20-03218-f010:**
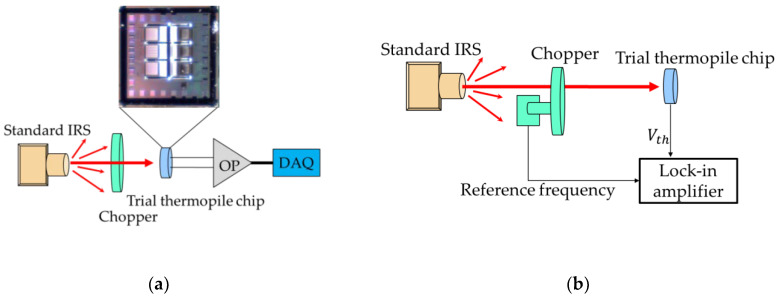
Setups of the experimental measurement framework of the output voltage (**a**) and the frequency response (**b**) for the trial thermopiles.

**Figure 11 sensors-20-03218-f011:**
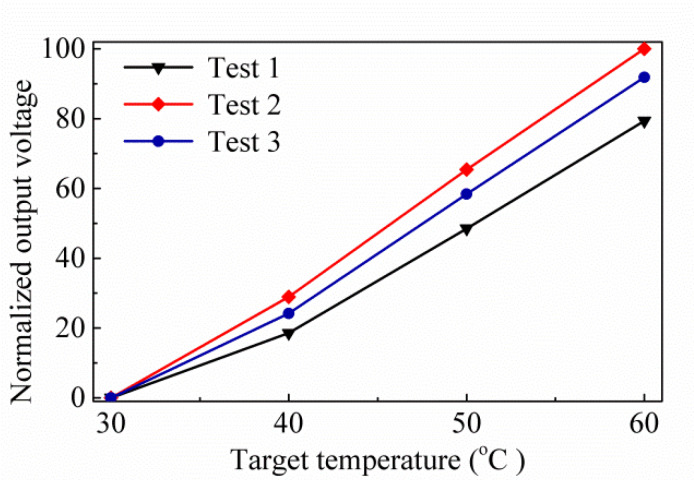
Normalized output voltages normalized by the maximum of test 2 case for three trial thermopiles.

**Table 1 sensors-20-03218-t001:** Range and interval of geometric parameters selected during simulation.

Geometric Parameters	Range (μm)	Interval (μm)
*ρ*	2.5–3.5	0.5
*w_x_*	3–21	3
*w_y_*	3–7	1

**Table 2 sensors-20-03218-t002:** Rectangular-hole sizes of three proposed SA type patterns with *ρ* = 2.5 μm for the trial thermopiles, their simulated IRAs, and normalized IAEs.

Test	Rectangular-Hole Size (μm)		Simulated IRA (%)	Normalized IAE (%)
	w_x_	w_y_		
1	12.5	5.5	85.57	88.9
2	15.5	5.5	94.39	100
3	18.5	5.5	91.25	94.76

**Table 3 sensors-20-03218-t003:** Measurement results for the geometric parameters of three proposed thermopiles.

Test	Geometric Parameters (μm)			
	*ρ_x_*	*ρ_y_*	*w_x_*	*w_y_*
1	2.4	2.53	12.60	5.47
2	2.4	2.49	15.60	5.51
3	2.39	2.51	18.61	5.49

**Table 4 sensors-20-03218-t004:** Simulation results of the normalized IAE normalized by the maximum of them at target temperature 60 °C for several thermopiles with SRHA of the given geometric parameters and without structure.

Arrangement	Geometric Parameters (μm)			Normalized IAE (%)
	*ρ_x_*/*ρ_y_* or *ρ*	*w_x_*	*w_y_*	
SA	2.4/2.49	15.60	5.51	100
MA	2.5	15.5	5.5	80.56
MA	3.5	15	3.5	68.08
Without Structure				30.03

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
