# Peer review of "Enhanced Infrared Absorbance of the CMOS Compatible Thermopile by the Subwavelength Rectangular-Hole Arrays"

_sensors, 2020, doi:10.3390/s20113218_

Round 1

Reviewer 1 Report

This manuscript describes a numerical and experimental study to improve the infrared absorption efficiency of CMOS compatible thermopiles using subwavelength rectangular hole arrays.  Several arrays were first investigated numerically using the finite difference-time domain method.  Based on this information, several test cases were manufactured and experimentally tested yielding good agreement with the numerical studies and improved absorption efficiency.   I thought the manuscript was well-presented and recommend publishing with a few minor changes listed below.   In the manuscript, the acronym IAE is listed without definition in the abstract--just simply define it in the previous line where
infrared absorption efficiency is noted.
  Line 35:  Currently reads "Thermocouple is a widely..."
I would suggest changing to "The thermocouple is a widely..."  or simply "Thermocouples are a widely..."
  Line 63:  Currently reads "In this study we are numerically and experimentally investigated for the enhanced..."
I would reword to something like, "In this study we numerically and experimentally investigated the enhanced..."
  Line 88:  Regarding the PML boundary condition.  I would suggest adding just a little more description of this boundary condition.   Section from 101-112:  I thought Figures 1 and 2 were very helpful in visualizing the numerical/experimental setup--but I had a hard time understanding what was being varied in the text in the series of lines from ~101-112.  I would recommend rewriting this section to improve clarity--perhaps making more references to Figures 1-2 to help the reader understand what is being done.

Author Response

Dear Editor:

Manuscript ID: sensors-819807

Type of manuscript: Letter

Title: Enhanced Infrared Absorbance of the CMOS Compatible Thermopile by the Subwavelength Rectangular-Hole Arrays

Authors: Chi-Feng Chen, Chih-Hsiung Shen *, Yun-Ying Yeh

Thank you for your letter dated May 27, 2020.

Thank you for your time and consideration of our work. And thank the referees for a thorough and helpful analysis of our manuscript. We have individually responded to all suggestions and comments from the referees and revised the manuscript accordingly.

  • Responses to Referee Comments:
  • Response to Anonymous Referee 1

Thank you for your valuable suggestions and comments. We have revised the manuscript accordingly and have detailed corrections and explanations point by point below.

  1. In the manuscript, the acronym IAE is listed experimentally investigated the enhanced infrared absorbance."

[Response] We fully agree and thank you for your suggestion. We have modified the manuscript.

  1. Line 88:  Regarding the PML boundary condition.  I would suggest adding just a little more description of this boundary condition.   

[Response] We have added several sentences to describe this boundary condition in the last line on page 5.

  1. Section from 101-112:  I thought Figures 1 and 2 were very helpful in visualizing the numerical/experimental setup--but I had a hard time understanding what was being varied in the text in the series of lines from ~101-112.  I would recommend rewriting this section to improve clarity--perhaps making more references to Figures 1-2 to help the reader understand what is being done.

[Response] We have referred your suggestion and modified for the sentences in the text in the 74-93 and 101-112 series of lines. We have re-described those sentences in modified manuscript in the 121-136 series of lines. We have added a table (Table 1) to describe the range of geometric parameters selected during simulation. And we have adjusted original Figures 2-3, 5-6, 8, and 13, and removed original Figure 4.

  • Response to Anonymous Referee 2

We are very sorry, our previous manuscripts are not appropriate for our results, causing you many misunderstandings. In order to avoid misunderstanding, we have modified for the sentences in the text in the 74-93 and 101-112 series of lines and re-described those sentences in modified manuscript in the 74-136 series of lines. We have added a table (Table 1) to describe the range of geometric parameters selected during simulation. And we have adjusted original Figures 2-3, 5-6, 8, and 13, and removed original Figure 4.

  1. What is the material of the active area? What is its original absorbance when there are no subwavelength structures?

[Response] A. In this study, the material of the active area is SiO2. To clearly present this related content, we have added some sentences in modified manuscript in the 84-105 and modified the relative annotation in Fig. 2.

  1. The original absorbance means the absorbance of thermopile without subwavelength structures in the active area.
  2. Can the authors compare the SA and MA absorbing areas with an active area without structures?

[Response] We are very sorry that our previous manuscript did not clearly show the relevant content, which caused your misunderstanding. SA and MA represent staggered arrangement of hole array and matrix arrangement of hole array, respectively. Because there are no SA and MA absorption regions, we cannot compare SA and MA absorption regions with effective regions without structures.

  1. As was stated by the authors the array thickness is 4um, to deposit the medium layer, what kind process the authors used? what was the temperature?

[Response] Because our words are not suitable, it caused your misunderstanding, the array thickness t shown in sub-graphs of Fig.1 means the wall width between two holes. Thanks for your suggesting, it is indeed easy to misunderstand. In order to avoid misunderstanding, we have added some words to explain its meaning, and modified the representative words, using “hole-wall width” to represent. The hole pattern is formed by Reactive Ion Etching (RIE) to remove SiO2 dielectric material on the absorption region in the RLS process. Next, the RLSSI process is used to further perform RIE etching of the silicon substrate beneath under the sensing structure. After the processes, the active area is floated and filled with an array of etched holes as a periodic index waveguide. After removing the silicon substrate under the cantilever, the SiO2 thin film structure in the active area has a thickness of 7 μm. Here RLS and RLSSI processes provided by TSRI(Taiwan Semiconductor Research Institute). In addition, to avoid misunderstanding, we have modified for the sentences in the text in the 74-93 and 101-112 series of lines and re-described those sentences in modified manuscript in the 197-220 series of lines. In addition, In TSMC 0.35 μm CMOS process, the medium layer is deposited and formed at the procedure of PECVD SiO2 and the process temperature is around 450 °C.

  1. When the width is 3um or 3.5um, the corresponding other width of the hole is 15um. What is the corresponding other width?

[Response] Since the rectangular hole has a width of two dimensions, “the corresponding other width” refers to the width of another dimension other than the width of the rectangular hole that has been expressed. In order to avoid misunderstanding, we have modified for the sentences in the text in the 74-93 and 101-112 series of lines and re-described those sentences in modified manuscript in the 121-136 series of lines.

  1. The reviewer cannot understand why the authors normalized the data with the maximum of them at 75 ° What is the IAE and output voltage at 75 °C?

[Response] The data is normalized by the maximum of them at 75 °C because the results obtained from the simulation and the results obtained from the measurement cannot be directly compared. So we normalize them individually using their maximum. We hope to use these individual normalized values for objective comparison. The IAE represent the infrared absorption efficiency obtained from the simulation and output voltage is obtained from the measurement. The IAE and output voltage at 75 °C is because we want to express that if there is a standard infrared radiation source, this source is set to 75 °C, it will radiate the radiation energy of 75 °C, and the CMOS compatible thermopiles with SRHA will be used to numerically and experimentally detect the radiation energy emitted by this standard infrared radiation source.

  1. As the author stated the IR absorbances are 78.23% and 83.66% for the two cases, what is the temperature?

[Response] We consider the wavelength range of the infrared radiation radiated by the target temperature of the radiation source. In this study, t the target temperature is set in the range of 30 degrees to 100 degrees, so the calculated wavelength range is correspondingly selected in the range of 8 μm to 10 μm. That is, the IR absorbance is calculated for the wavelength range of 8 μm to 10 μm. We have added the two sentences to explain it in the modified manuscript in the  139-142  series of lines.

  1. When the array thickness becomes smaller, the absorbance becomes larger. What is the reason?

[Response] First, let ’s first explain that, as described above, the array thickness means the wall width between two holes. The detailed physical mechanism is not yet clear, but the simulation results show that when such thermopile has the subwavelength structures in the active area, the reflected light wave received by the reflection detector shown in Fig. 2 will be reduced, and more light waves can be transmitted into the active area. In short, when the geometric size of the subwavelength holes is not suitable, the total the reflection part and the transmission part of radiant energy respectively received by the reflection detector and the transmission detector will increase, that is, the absorption of active area medium becomes smaller. Therefore, according to comparing the presence and absence of the subwavelength holes, the energy of the reflected light wave will change significantly. We believe that the decrease in reflectance is one of the reasons for the increase in the absorption of active area medium.

  1. In Figure 13, what are the testing samples from Test 1 to Test 3?

[Response] For the testing samples from Test 1 to Test 3, the data is list in Table 2 in modified manuscript.

  1. The manuscript is too long, with too many figures.

[Response] For the manuscript is too long, we have modified the manuscript to make it more concise and clear. For too many figures, we have removed original Figures 4 and 10, and combined original Figures 11-12 into the modified Figure 10 (a) and (b).

Thank you and best wishes,

Chi-Feng Chen

Department of Mechanical Engineering/Institute of Opto-Mechatronics Engineering,
National Central University,
No.300, Jhongda Rd., Jhongli City, Taoyuan County 320, Taiwan(R.O.C.)
Phone: +886-3-4267308
Fax: +886-3-4254501
E-mail: [email protected]

Reviewer 2 Report

  1. What is the material of the active area? What is its original absorbance when there are no subwavelength structures?
  2. Can the authors compare the SA and MA absorbing areas with an active area without structures?
  3. As was stated by the authors the array thickness is 4um, to deposit the medium layer, what kind process the authors used? what was the temperature?
  4. When the width is 3um or 3.5um, the corresponding other width of the hole is 15um. What is the corresponding other width?
  5. The reviewer cannot understand why the authors normalized the data with the maximum of them at 75 oC. What is the IAE and output voltage at 75 oC?
  6. As the author stated the IR absorbances are 78.23% and 83.66% for the two cases, what is the temperature?
  7. When the array thickness becomes smaller, the absorbance becomes larger. What is the reason?
  8. In Figure 13, what are the testing samples from Test 1 to Test 3?

9. The manuscript is too long, with too many figures.

Author Response

(The authors gave the same response as above.)
